# Evaluation of Genotoxicity and Toxicity of *Annona muricata* L. Seeds and In Silico Studies

**DOI:** 10.3390/molecules28010231

**Published:** 2022-12-27

**Authors:** Gleison Gonçalves Ferreira, Ana Carolina Sousa Quaresma, Dayse Lúcia do Nascimento Brandão, Andrey Moacir do Rosario Marinho, José Edson de Sousa Siqueira, Kamila Leal Correa, José Otávio Carréra Silva-Júnior, Sandro Percario, Maria Fâni Dolabela

**Affiliations:** 1Faculty of Pharmacy, Federal University of Pará, Belém 66075-110, Brazil; 2Postgraduate Program in Pharmaceutical Sciences, Institute of Health Sciences, Federal University of Pará, Belém 66075-110, Brazil; 3Health Department of Pará, Castanhal 66093-677, Brazil; 4Institute of Exact and Natural Sciences, Federal University of Pará, Belém 66075-110, Brazil; 5Institute of Biological Sciences, Federal University of Pará, Belém 66075-110, Brazil

**Keywords:** medicinal plants, cancer, *Allium cepa*, *Artemia salina*, in silico

## Abstract

Cancer is a multifactorial organic dysfunction for which great efforts are being devoted in searching for new treatments and therapeutic adjuvants. *Annona muricata* is a fruit that has promising activity against several types of cancer, as it contains acetogenins, the metabolite group associated with this action. Thus, the objective of this study was to evaluate, in experimental models, the toxic behavior of an extract and fraction rich in acetogenins from *A. muricata* seeds and study the acetogenin, Annonacin, in silico. Phytochemical characterization was made by thin layer chromatography, spectroscopy in the infrared region and nuclear magnetic resonance. Toxicity was evaluated by tests of *Allium cepa* and *Artemia salina*, and in silico studies using the SwissDock servers DockThor, PharmMapper, ADMETLab, PreADME, Osiris and ProTox. The extract and fraction showed genotoxic activity against meristematic cells of *A. cepa*, reducing the mitotic index; however, the extract produced great deleterious effects on the system, even causing cell necrosis. In *A. Saline,* the extract was more toxic than the fraction, but both samples were considered toxic. Annonacin was effectively linked to complex I, and presented different activities regarding toxicity. Thus, the results of this study are promising, highlighting the anticancer potential of acetogenins.

## 1. Introduction

Cancer is a disease defined by abnormal cell proliferation resulting from multiple factors. The National Cancer Institute (INCA) estimates that between 2020 and 2022, there will be 625,000 new cases of cancer (non-melanoma skin). Conventional antineoplastic therapies tend to present disadvantages in the treatment adhesion, toxicity, and general improvement of patients’ quality of life [1]. Research to develop or characterize new antineoplastic or adjuvant therapies are essential in terms of the aforementioned aspects. Therapeutic searches using plant species can be important in developing new medicines. Phytochemical analyzes elucidate new compounds that produce desired therapeutic actions, opening a range of possibilities [2].

*Annona muricata* is a plant belonging to the Annonaceae family of the Magnoliales order. It has great economic and cultural value, and is used in various culinary preparations. In the pharmacological field, its medicinal use is highly widespread in Central and South America, as well as countries in Africa and Asia. Among its popular uses are treatments for bone disorders, neuralgia, digestive disorders, febrile illnesses, abrasions, antiparasitic, cystitis, diabetes, headaches, insomnia, abscesses, ulcers caused by leishmaniasis, conditions associated with cancer, and as an insecticide, larvicidal and piscicidal agent [3].

In vivo studies have shown that *A. muricata* leaf extracts have an anti-rheumatic effect by decreasing the production of pro-inflammatory cytokines. Another study reached the same result using fruits with antinoceptive activity that results from a decrease of mediators and activity of the opioidergic pathway. Antiparasitic activity was demonstrated in vitro against *Trypanosoma cruzi* and against several strains of Leishmania, using, in the latter case, mainly isolated acetogenins. Among the bioactive compounds isolated from *A. muricata*, alkaloids and acetogenins stand out, and megastigmans, flavonoids, phenols and cyclopeptides can also be found [4].

As much as *A. muricata* is recognized for generating these numerous metabolite groups, in recent years, studies have shifted from alkaloids to acetogenins. This highly bioactive group includes compounds obtained by the acetic acid-polyketide derivatives of long-chain fatty acids. In addition, these aliphatic chains have functional groups attached to them, such as hydroxyl, acetyl and carbonyl groups, as well as a terminal *y*-lactone ring, tetrahydrofuran, or tetrahydropyran rings. The structural organization of these functional groups has been used to establish the cytotoxic potential of acetogenins. The hydroxyl groups, the presence of the THF ring, and α,β-unsaturated y-lactone subunit, are essential to produce the toxic effect of these compounds, in addition to their concentration in the extract [3].

Studies using extracts from different parts of *Annona muricata* showed that the fruit has targeted activity against different neoplasms, with acetogenins the main group responsible for this activity. The ability of these secondary metabolites to produce cytotoxic effects in carcinogenic cells is mainly, but not exclusively, linked to their ability to inhibit mitochondrial Complex I [5], which is the enzyme responsible for initiating the process of cellular respiration. This inhibition may lead to cell death of neoplastic cells [6,7].

Studies that assess the toxic capacity of a compound, including cytotoxicity and genotoxicity studies, are essential to determine the safety profile and future uses of the compound via its pharmacological action. The in vitro and in vivo evaluation of extracts or isolates from plant species has become a powerful sieve in the selection of bioactive agents capable of becoming new drugs. In addition, in silico studies provide evidence that can be investigated through other experimental assays, which allows creating a "shortcut" that is based on a complex series of algorithms. Making small changes in the structure of these chemical compounds, making them more active, selective and less toxic, can lead to a reduction in the production costs of a new drug. This work investigates the genotoxic and toxic effects of an extract and a fraction obtained from *A. muricata* seeds, and includes an in silico study with Annonacin as a target [4].

## 2. Results and Discussion

### 2.1. Preparation of Extract and Fraction, and Phytochemical Characterization

After seed maceration, an oily extract with a mass of 5.28 g was obtained, which represented 17.6% of the initial mass of the extractive process. From this final mass, 3 g was used for liquid-liquid partition, 1.5 g in each and, after exploratory TLC, the dichloromethane fraction (FDSAM) was chosen for the following tests. FDSAM represented most of the mass after partition, and 0.76 g was used in the analysis as a fraction rich in acetogenins.

TLC chromatographic profiles suggested that acetogenins were obtained for both the EESAM (ethanolic extract) and its fraction; however, when observed under ultraviolet light, profiles indicative of alkaloids were also noted (Figure 1). Both alkaloids and acetogenins act as phytochemical markers in Annonaceae. Portion A showed a chromatoplac under ultraviolet light (365 nm), the fluorescent blue bands being suggestive of alkaloids, which was confirmed by the reaction with Dragendorff, as demonstrated in portion C. Portion B demonstrates the presence of acetogenins by the reaction with Kedde, which generates a color between rose and lilac, usually weak, that dissipates in a few seconds, revealing acetogenins that have α, β-unsaturated γ-lactonic ring. As observed in portions B and C, acetogenins and alkaloids appear at the same point of retention, which may indicate there is a mixture of the two metabolite groups. To determine if FDSAM is rich in acetogenins, analyzes were performed by more sensitive methods.

We observe the presence of C=O deformations in the infrared spectrum (Figure 2) (1745 and 1653 cm^−1^), indicating the lactone portion of acetogenins, and we also observed the axial deformation of C=C (1458 and 1465 cm^−1^) and the axial deformation C-O (1060 cm^−1^) that suggests the portion connected to the THF ring (tetrahydrofuran), while the deformations C-H (3003 cm^−1^), OH (3338 cm^−1^), CH2 (2850 cm^−1^) and CH3 (2920 cm^−1^) indicated the elongated chain of acetogenins linked to the lactone ring and the THF ring. Thus, the absorptions described here indicate the presence of long chains of fatty acids that point to the presence of acetogenins [8,9].

In the ^1^H NMR spectra for the two samples, we found signals of acetogenins, where the α,β-unsaturated *y*-lactone subunit was evidenced by δ 1.4; 5.0 and 7.0 or 7.2 for acetogenins with OH in their chain, for carbons linked to hydroxyls in shifts δ 3.0 and signals linked to the presence of the THF ring δ 3.8. We also observed characteristic signals of H linked to olefinic carbons at δ 5.0, as well as a large amount of δ 2.0 shifts that are related both to the lactone moiety and to chemical shifts of hydrogens present in the acetogenins chain (Figure 3 and Figure 4; Table 1) [3,10].

Thus, the extract and the fraction showed the presence of acetogenins in the three tests performed. Studies with *A. muricata* seeds have generally focused on the presence of acetogenins; however, phytochemical surveys of leaves have demonstrated the presence of a range of other metabolite groups, such as tannins, flavonoids, triterpenes and alkaloids [11]. In our study with seeds, alkaloids and acetogenins were present in this part of the plant, and could be used as pharmacogens in future studies.

### 2.2. Genotoxicity in Allium cepa

The experimental model of *Allium cepa* allows the evaluation of the genotoxic effects of organic and inorganic compounds from alterations in mitotic index (MI) and the presence of alterations in the mitotic cycle, multinucleated cells or micronuclei, formation of bridges (linked nuclei), index of aberrations (IA), these being cheap and simple tests [12]. The results generated by such tests, as in this study, allow us to understand the mechanism of cytotoxic effects already described for extracts of *A. muricata*.

In this study, the extract and fraction demonstrated genotoxic and cytotoxic actions by decreasing MI and increasing IA. EESAM and FDSAM showed, at the lowest concentrations, an elevation of the MI that decreased as the concentration increased and the time elapsed, which may indicate they are time-dose dependent compounds. The elevation of MI compared to the positive control indicates a cytotoxic effect based on promotion of cell growth and eventual tumorization, while a decrease of MI indicates antiproliferative characteristics of the compounds [13], as observed samples in Table 2, which tend to decrease cell proliferation.

Other studies using *A. cepa*, showed that the extracts of *A. muricata* have a cytotoxic effect; however, many of these studies used the leaves or bark [14,15,16,17,18,19], and only in a few studies were seeds and fractions rich in acetogenins used, which, in recent years, have been associated with the antitumor effects of *A. muricata* [20].

Studies have demonstrated the cytotoxic activity of acetogenins isolated from *A. muricata* against tumor strains demonstrated with IC_50_ values ranging from 8 × 10^−3^–3.3 × 10^−1^ µg/mL, i.e., against A-549 (human lung carcinoma) find activities for Annomurin with A, B, C and E; Annomutacin, Annopentocin with A, B and C; Arianacin, cis-annonacin, cis-annonacin-10-one, cis-goniothalamicin, cis-trans-annomuricin-D-one, Goniotha-lamycin, Javoricin, Muricatetrocin C, Muricatocin with A, B and C; and Muricoreacin and Murihexocin with A, B and C. In MCF-7 (mammary carcinoma) the same acetogenins added to 2.4-cis and trans-10R-annonacin-A-one presented IC_50_ values between 5.70 × 10^−2^ and 17.93 µg/mL. Against HT-29 (human colon adenocarcinoma), the same isolates showed an IC_50_ of 9 × 10^−4^–4.0 µg/mL. IC_50_ values ranged from 1.71 × 10^−2^–1.14 µg/mL for PC-3 (prostate adenocarcinoma) Annomuricin with E, and Annopentocin with A, B and C; and for cis-trans-annomuricin-D-one, Muricapentocin, Muricoreacin, and Murihexocin with A, B and C [21].

When comparing the results, there was no statistical difference between the positive control and the samples, demonstrating the similarity between the toxic effects of colchicine, EESAM and FDSAM. Colchicine has been evaluated as an inducer of chromosomal alterations due to meiotic changes, mainly in chromosomal adherence and lag, and abnormalities related to mitotic spindle dysfunction [22].

Studies that evaluated the mechanism of action by which acetogenins exert their cytotoxic activity pointed to an inhibition of the NADH-ubiquinone oxidoreductase complex, where acetogenins act by inhibiting electron transport by blocking complex I [23,24]. The electron transport chain produces energy for cells by the oxidative phosphorylation of glucose and fatty acids. Complex I reduces NADH to NAD+, in addition to ubiquinone to ubiquinol, and carries protons through the inner mitochondrial membrane through an imbalance of the redox potential. In this way, cells that require a large amount of energy for division, such as neoplastic cells, would be strongly affected by this effect [5,25].

As observed by this study, EESAM and FDSAM produced deleterious effects on cells, leading to visible damage in the cellular environment. The extract and the fraction showed vagrant chromosomes, chromosomal loss, telophase bridges and micronuclei (Figure 5 and Figure 6), which indicates there were alterations in the mitotic process. EESAM produced harmful effects on *A. cepa* meristem cells such as loss of shape, chromosomal viscosity and necrosis. In this sense, the fractionation process maintained the cytotoxic effects without damage caused by the extract that could come from a synergistic effect with other, non-alkaloid compounds existing in the seed.

### 2.3. Toxicity in the Artemia salina Leach

Seeking to expand the results discussed in this study, the toxicity test was performed against *Artemia salina*. The same concentrations used in the genotoxicity test were used to observe how the samples would behave in a more complex organism. EESAM had an LC_50_ value of 1.26 µg/mL, while that of FDSAM was LC_50_ = 4.58 µg/mL (Figure 7). In the consulted literature, there were no studies that used extracts or fractions of seeds in concentrations close to those of our study. However, Luna et al. [26] found CL_50_ = 0.49 µg mL^−1^ for leaf extracts. Another study showed the higher the concentration, the higher the percentage of deaths, thus demonstrating a concentration-dependent action [27]. Studies by Hoe et al. [28] showed that extracts rich in acetogenins are more toxic against *A. salina* larvae, confirming the results presented in this study.

### 2.4. In Silico Study

In silico studies have emerged as a prerequisite for the development of new drugs or analyzes of molecular changes made to existing drugs. In this study, we studied Annonacin, an acetogenin isolated from the seeds of *A. muricata*, as a target. This molecule was chosen because it shows signals in the infrared spectrum and the ^1^H NMR method used in this study [3,10]

When performing molecular anchoring targeting with Complex I (Figure 7), we obtained an affinity of −10.72 Kcal/mol (T. Energy −54.543 Kcal/mol) on SwissDock and −8.13 Kcal/mol (T. Energy −64.897 Kcal/mol) on the DockThor. The activity of a molecule is related to the energy it gives off to bind to a target. In this relationship, a more stable complex requires less energy for the interaction to occur [28]. Thus, the negative values found in this study demonstrate Annonacin, theoretically, effectively binds to mitochondrial Complex I, according to our hypothesis.

By simulating the bonds that Annonacin can make with Complex I (Figure 8) we obtained evidence for the presence of two hydrogen bonds, one conventional with a Glycine-Gly89 residue (Å 3.03), and another hydrogen pi-bridge with Fenylalamine-Phe93 (Å 3.09) and two hydrophobic alkyl ones with residues Alanina-Ala91 (Å 4.47) and Ala426 (Å 4.04) (Figure 9). The smaller the distance between the amino acid residues and the ligand, the stronger the bond, even if these bonds are weak, as with hydrogen bonds [29]. The distances observed in this study, up to 4.5 Å, indicate good coupling between Annonacin and Complex I [24].

When generating reverse pharmacophoric mapping, we obtained 300 possible targets for Annonacin. The first 10 were studied in more detail and are presented in Table 3, with emphasis on GTPase HRAS, transthyretin, mitogen-activated protein kinase and methionine aminopeptidase 2, which were chosen because they relate, directly or indirectly, to carcinogenic processes or/and tumor deactivation, which was the subject of this in silico study.

The first predicted target was the GTPase HRAS, an isoform of the GTPase RAS protein expressed in several types of cancer. RAS proteins are considered “undruggable” because, in addition to not having favorable binding sites, they make high-affinity bonds with nucleotides. Thus, drugs that seek to inhibit this pathway competitively inhibit the docking of RAS-bound GTP to the membrane. Drugs that block GTP are prime targets in the search for new antineoplastic therapies [30,31].

The second target was Transthyretin (TTR). Another protein involved in the development of cancer, it is one of the biomarkers of cell proliferation. In lung cancer, evidence was found that recombinant TTR increased the proliferation of carcinogenic cells acting by several mechanisms, such as increased CLL melanoma cells, regulation of immune cells and promotion of angiogenesis. In addition, it acts as a carrier of cellular retinoic acid binding protein (third predicted target) [32,33]. Thus, drugs that modulate TTR can be used in specific cases of lung cancer.

The seventh predicted target was the mitogen-activated protein kinase (MAPK) family, specifically p38 MAP kinase, which shows dual characteristics in cancer pathogenesis. p38 acts both in the G1/S and G2/M phases by checkpoint responses. As an antitumor, p38 has mechanisms of negative regulation of Cyclin D1 and, through cellular senescence, as a pro-oncogenesis, it increases resistance, survival and even assists migration (metastasis) of cancer cells from the isoform p38γ [34,35]. Therefore, they may be targets of new therapies, as long as they are selective for the p38γ isoform.

Methionine aminopeptidase 2 (MetAP2) was the eighth target predicted in this study, and is related to tumor activities. MetAP2 is a bifunctional enzyme that removes the amino-terminal methionine from newly formed proteins. MetAP2 seems to be related to the proliferation of endothelial cells at the stage of tumor angiogenesis, and has thus become a popular target for antitumor drugs [36,37].

Table 4 shows toxicity predictions for Annonacin in different servers. We found resonances between the predictions made for hepatotoxicity, hERG inhibition (cardiotoxicity), irritant effects and mutagenicity; however, we found dissonant results for carcinogenesis. While Osiris and ProTox showed negative results, PreADME showed positivity in mice, which were more sensitive models than rats, in which a range of toxic substances were generated and resulted in a prediction of carcinogenicity [38]. PreADME was also predicted to be toxic to algae, Medaka, Daphnia and Minnow; however, highly lipophilic compounds tend to be predicted as toxic in these models because of their natural accumulation [39]. Studies using Annonacin focused mainly on evaluating its mutagenic action, being negative in these studies [40,41].

Studies have shown that Annonacin can have a neurotoxic effect, inducing an atypical Parkinsonian state. This could occur due to these compounds being deposited in cells of the nervous system, delays in the mitochondrial distribution of ATPs in the cell soma and interference in the intracellular distribution of the protein tau [42,43,44]. The authors of the studies were not categorical regarding this action, and there may even be synergism for the toxic effect [42]. In addition, preliminary studies have demonstrated that even though acetogeins, such as Annonacin, can cross the blood-brain barrier, there is low brain exposure to acetogenins at doses of 10 mg/kg [45].

Thus, what is envisaged is a compound with high potential for use against neoplasms. However, more detailed studies are needed not only on the pharmacokinetics of this compound, but also on the pre-formulation of its possible pharmaceutical form, to guarantee its efficacy and safety as a therapy, as well as molecular modeling studies in which its therapeutic effect can be guaranteed, reducing its toxicity.

## 3. Materials and Methods

### 3.1. Plant Material, Extract and Fractions

*A. muricata* seeds were donated by Miriam Beta. They came from the municipality of Castanhal—PA. They were sanitized, dried and ground. An exsiccata was prepared and it is in the process of being integrated into the Herbarium Prof. Dr. Marlene Freitas da Silva—Center for Social Sciences, at the University of the State of Pará.

The ground seeds (±30 g) were subjected to exhaustive maceration with 96° GL ethanol for 21 days. Every 7 days the extractive solution was filtered, and fresh solvent was added. At the end of 21 days, the extractive solutions were concentrated in a rotary evaporator under reduced pressure, obtaining EESAM from the seeds. To obtain the acetogenin-rich fractions, liquid-liquid partitions were carried out, where 1.5 mg of EESAM was solubilized in methanol and water (1:1) and extracted with organic solvents, as in the scheme described in Figure 10.

### 3.2. Thin Layer Chromatography—TLC

Silica gel was used as the stationary phase, and 18 mL of ethyl acetate and 2 mL of methanol (9:1 ratio) as a mobile phase. The reagents were: Kedde (for solution A (3.5 dihydrobenzoic acid (2 g) was mixed in methanol (50 mL)); for solution B (potassium hydroxide (5.7 g) was dissolved in methanol (100 mL)) and solutions were mixed in a 1:1 ratio at the time of use) and Dragendorff (bismuth subnitrate (8 g) potassium iodide (27 g) was mixed with concentrated hydrochloric acid (20 mL) and distilled water qsp. (100 mL)) [3].

### 3.3. Spectroscopy in the Infrared Region—Fourier Transform (FT-IR)

Spectra were obtained using an IR Prestige-21 Shimadzu spectrophotometer, in which 1 mL of the samples was dispersed over a diamond crystal with attenuated total reflectance (ATR), and 20 scans were performed with a resolution of 2 cm^−1^ in the spectral range of 4000–400 cm^−1^.

### 3.4. ^1^H Nuclear Magnetic Resonance

NMR analyzes were performed on a Bruker Ascend 400 spectrometer (operating at 400 MHz for hydrogen). Samples were solubilized in chloroform. Chemical shifts (δ) were measured in ppm and coupling constants (J) in Hertz (Hz). Tetramethylsilane (TMS) was used as an internal reference.

### 3.5. Allium cepa Assay

About 1 mg of *A. cepa* seeds was placed in a petri dish lined with filter paper and treated with different concentrations of ethanol extract and dichloromethane fraction from *A. muricata* seeds (25; 12.5; 6.25; 3.125 and 1.5625 µg/mL). After radial growth of 1 cm of the seeds, they were removed at 24, 48 and 72 h and were counted and fixed in Carnoy (absolute ethyl alcohol: glacial acetic acid 3:1).

For the control groups, distilled water (negative control) and colchicine (positive control) at the same concentrations of the extract and fraction were used. To prepare the slides, the shoots were removed from the fixative and transferred to a petri dish where they were washed (3 washes of 5 min). After that, they were subjected to acid hydrolysis in 1N hydrochloric acid for 15 min, then washed again and stained in acetic acid orcein 2% for 10 min. Following this, the shoots were placed on a slide and a drop of acetic orcein was placed on the cut. After 1 min, crushing (squash) was performed, and the slides were observed under the light microscope at 400× magnification [12]. The results were analyzed with the PAST program (Paleontological Statistics) using ANOVA followed by Tukey’s test for multiple comparisons. Values in which the p-value was less than or equal to 0.05 were considered statistically significant.

### 3.6. Artemia Salina Leach Assay

To hatch *A. salina* cysts, a solution of sea salt and distilled water at a concentration of 35 g/L was used. The pH of the solution was adjusted to 9–10 with sodium hydroxide (NaOH) at 0.1 mol/L, then the solution was homogenized and filtered. The saline environment was transferred to an aquarium with a light/dark environment, where 20 mg of *A. salina* cysts were added and remained until complete hatching cysts (±24 h) at a temperature of 27–30 °C. The extract and the fraction were prepared at the same concentration as the *A. cepa* assay with a final volume of 5 mL. This assay was carried out with five replicates, with 10 crustaceans being added to each and kept for 24 h and 48 h. The live and dead nauplius were counted at the end of each cycle [46,47] (both with adaptation). To calculate the Lethal Dose 50% (LD_50_) the IC_50_ Calculator tool [48] was used.

### 3.7. In Silico Study

The following programs were used in the in silico study: ChemSketch (version 2.1, 2019), to design the chemical structure of the Annonacin, which was subjected to molecular anchoring using the SwissDock (version 2022), and DockThor (version 2.0) against coenzyme NADH: ubiquinone oxidoreductase—Complex I (ID: 5XTB). These online servers were comprised within the fast-docking system. The Dockthor server was powered by the National Laboratory of Scientific Computing, which works with the C++ language, using generic algorithms based on the molecular force field MMFF94 (Merck Molecular Force Field). SwissDock, on the other hand, is a server that is confined to a system based on multiobjective scoring function designed around the CHARMM22 force field and the FACTS solvation model. The coenzyme and the ligand were optimized in the Discovery Studio program. In the program, the forces involved in the docking, receptor-ligand interactions, as distance and binding types, were observed. In addition to reverse pharmacophoric mapping using the PharmMapper server (2008–2022), PreADMET (2012–2021), ADMETLab (version 2.0) and Protox (2021), were used in addition to free software OSIRIS Property Explore (2017) [49,50,51,52,53].

## 4. Conclusions

Both EESAM and FDSAM showed genotoxic action against *A. cepa* and were toxic against *A. salina*; however, EESAM showed greater deleterious effects on cell division and microcrustaceans, which may result from synergistic action or the presence of specific toxicant groups. From the assays developed in this study, and the available literature, it is evident that acetogenins are a group of compounds that can be tested as drug candidates based on in vivo seminal studies and, in the future, may be used in broad or narrow antineoplastic therapy.

## Figures and Tables

**Figure 1 molecules-28-00231-f001:**
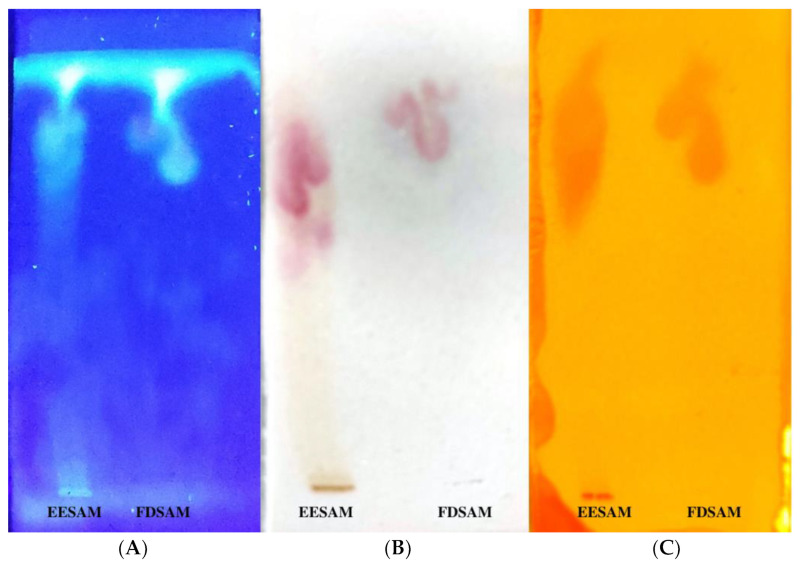
TLC chromatographic profile of the extract and fraction of *A. muricata*. EESAM—ethanolic extract from seeds; FDSAM—Dichlomethane fraction from seeds. Mobile phase: Ethyl acetate: Methanol (9:1). Developers: (**A**)—365 nm ultraviolet light; (**B**)—Kedde; (**C**)—Dragendorff. Source: authors.

**Figure 2 molecules-28-00231-f002:**
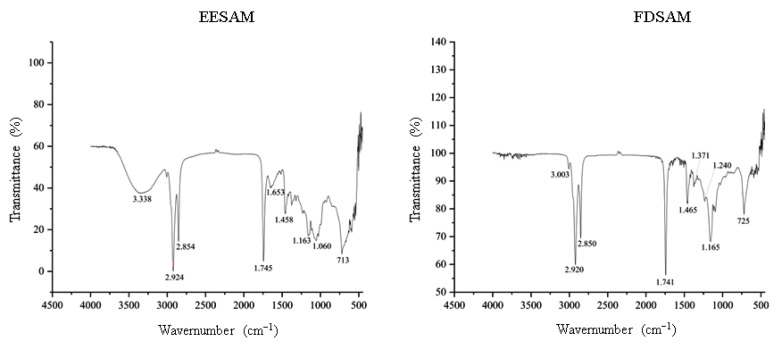
Infrared spectra of the extract and fraction of *A. muricata*. EESAM—ethanolic extract from seeds; FDSAM—dichlomethane fraction from seeds. Source: Authors.

**Figure 3 molecules-28-00231-f003:**
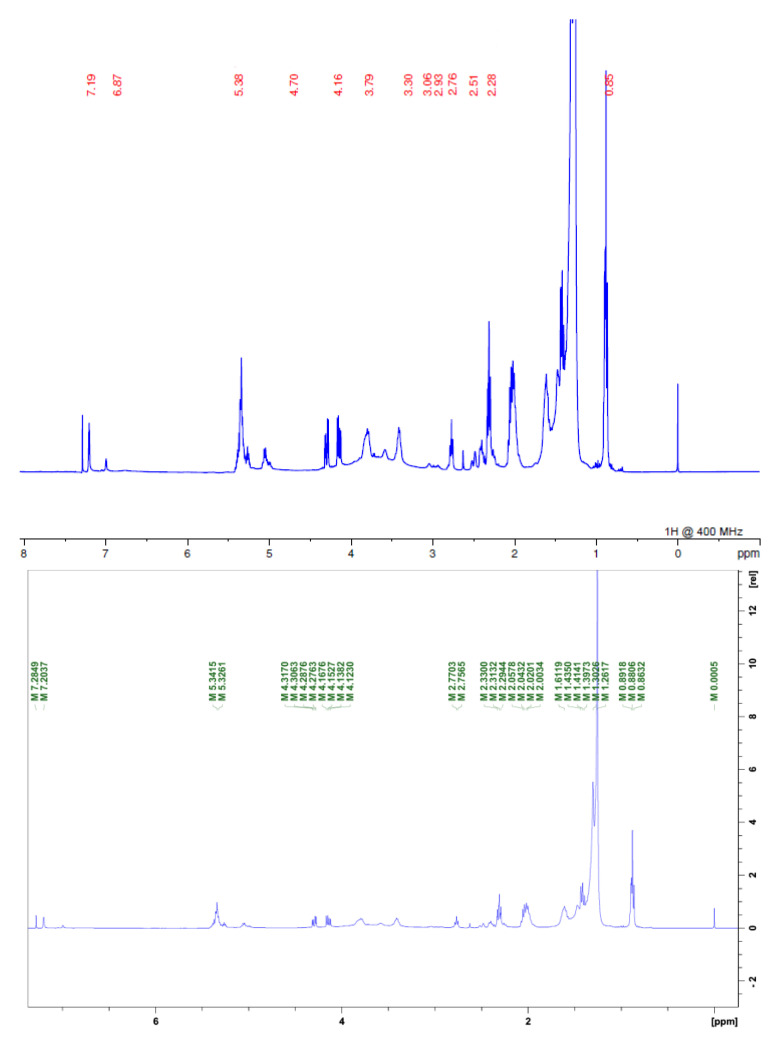
EESAM ^1^H NMR spectrum. Source: Authors.

**Figure 4 molecules-28-00231-f004:**
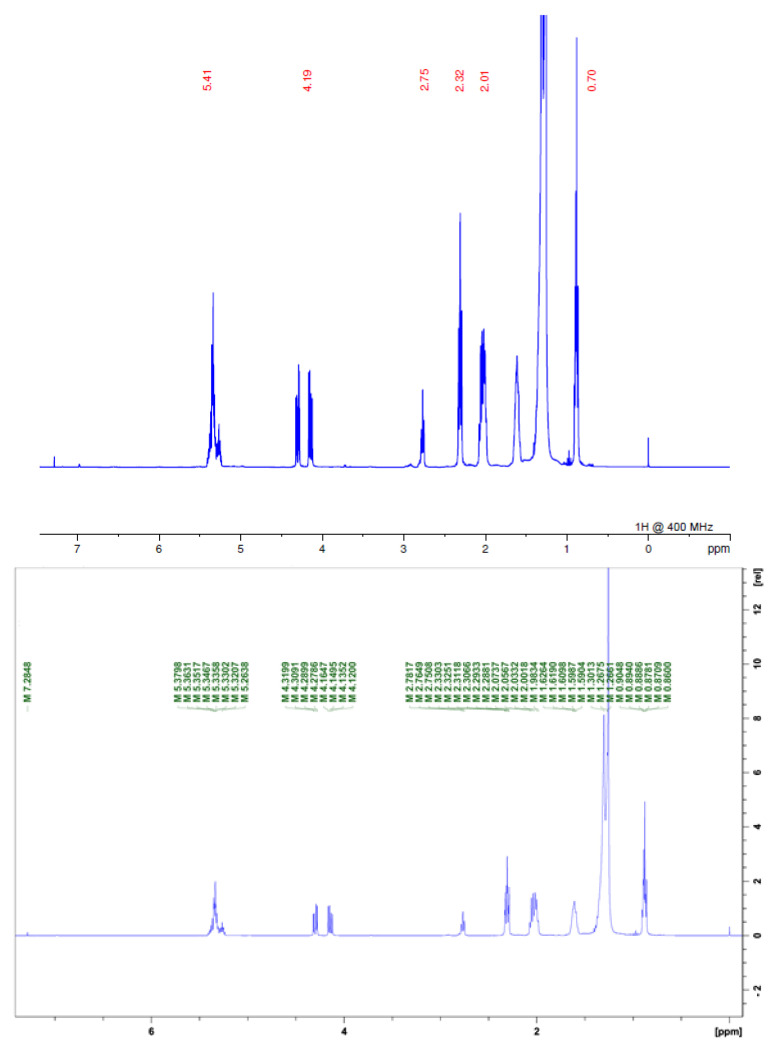
^1^H NMR spectrum of FDSAM. Source: Authors.

**Figure 5 molecules-28-00231-f005:**
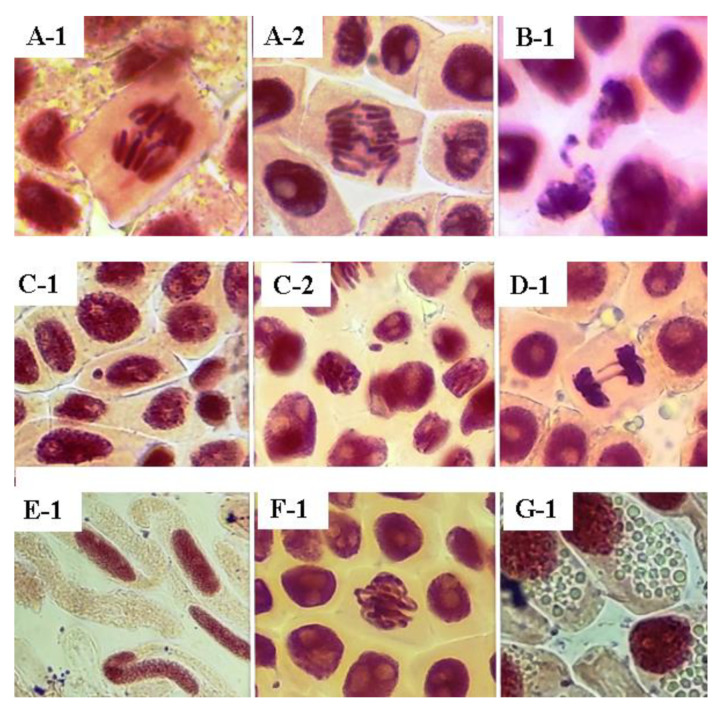
Chromosomal aberrations caused by the extract from *A. muricata* seeds. Source: Author, 2022. (**A-1**–**2**): Wandering chromosomes; (**B-1**): chromosomal loss; (**C-1**–**2**): micronuclei; (**D-1**): telophase bridge; (**E-1**): shape changes; (**F-1**): viscosity; (**G-1**): necrosis—detected by the presence of vacuoles in the cytoplasm.

**Figure 6 molecules-28-00231-f006:**
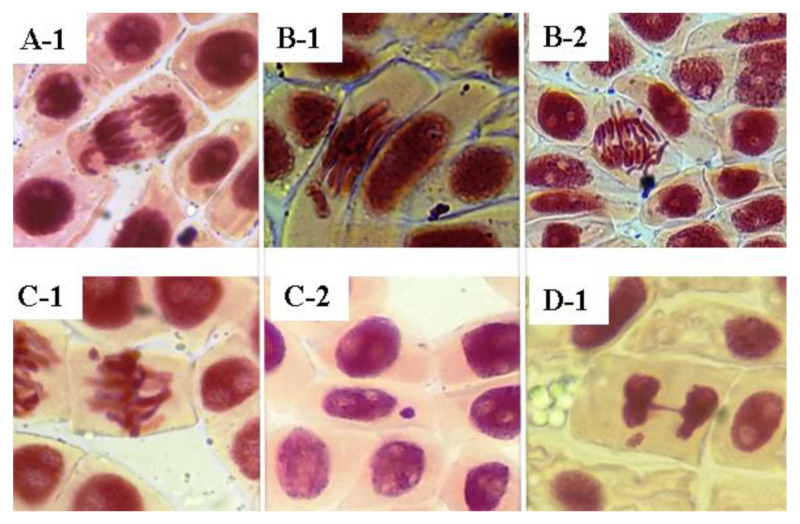
Chromosomal aberrations caused by the fraction from *A. muricata* seeds. Source: Author, 2022. (**A-1**): Wandering chromosomes; (**B-1**–**2**): chromosomal loss; (**C-1**–**2**): micronuclei; (**D-1**): telophase bridge.

**Figure 7 molecules-28-00231-f007:**
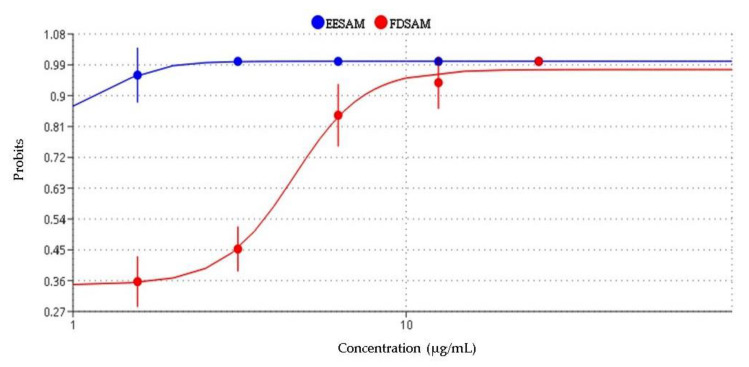
Determination of lethal dose in *A. salina* larvae. Source: Authors.

**Figure 8 molecules-28-00231-f008:**
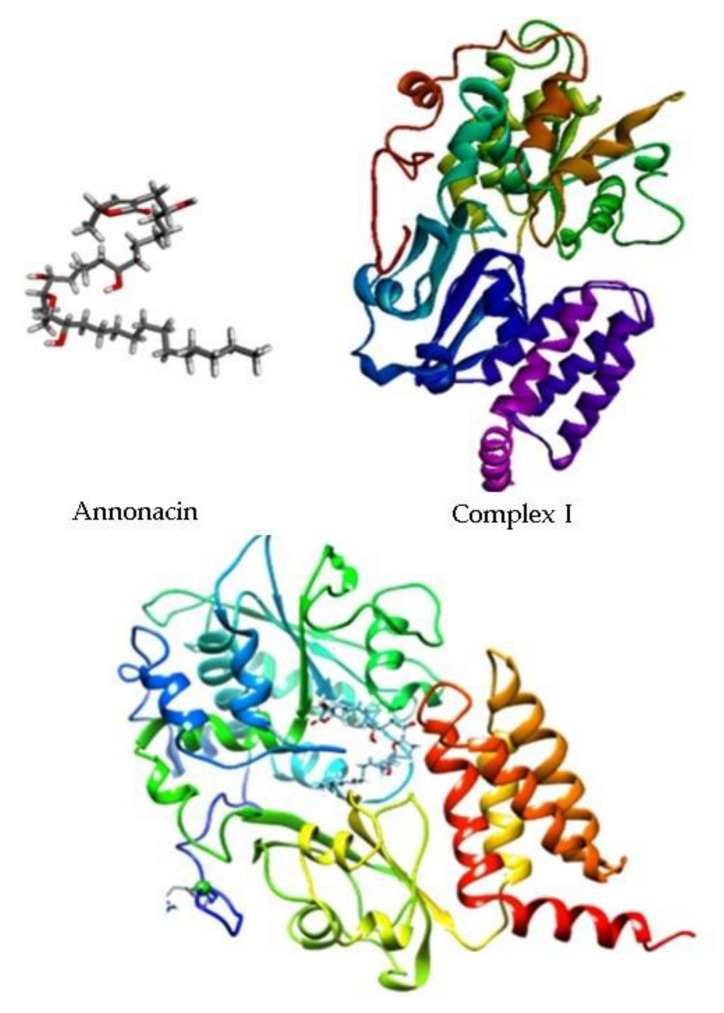
2D representation of molecular anchorage obtained by SwissDock. Source: Authors.

**Figure 9 molecules-28-00231-f009:**
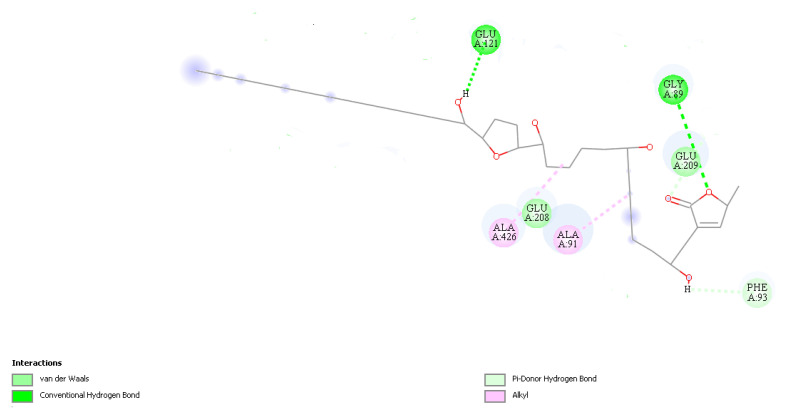
Simulation of connections made between the receptor-ligand. Source: Authors.

**Figure 10 molecules-28-00231-f010:**
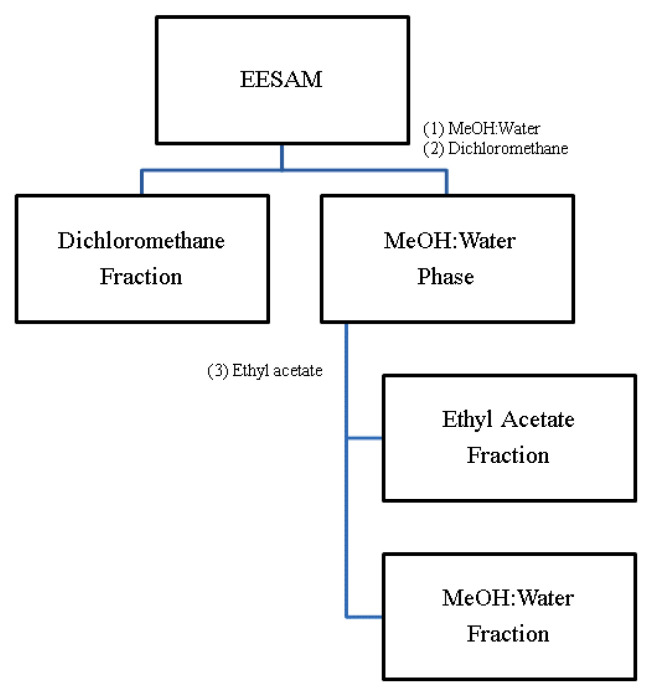
Proposed partition scheme. Source: Silva, 2017 (Adapted).

**Table 1 molecules-28-00231-t001:** Values of chemical shifts δ characteristics of Annonacin.

α,β-unsaturated *y*-lactone	H-2	H-3	H-4	H-35 (33)	H-36 (34)	H-37 (35)
	-	2.26	1.52	6.95	4.99	1.42
THF ring	δ					
	3.0–4.2					
Olefinic C-H	δ					
	5.0–5.8					

Source: [3].

**Table 2 molecules-28-00231-t002:** Mitotic index and cellular aberrations in *Allium cepa* induced by the extract and fraction of *Annona muricata*.

	Mitotic Index (MI) %			Chromosomal Aberrations (CA) %		
Sample	24 h	48 h	72 h	24 h	48 h	72 h
Distilled water (NC)	16.5	16.9	13.2	*	0.001	0.001
Colchicine (PC)						
1.5625 µg/mL	20.5	18.7	15.7	0.23	0.25	0.34
3.125 µg/mL	19.1	16.4	13.8	0.36	0.4	0.45
6.25 µg/mL	15.9	13.7	10.3	0.68	0.52	0.75
12.5 µg/mL	11.3	8.7	6.1	0.81	0.89	1.23
25 µg/mL	5.9	4.2	1.2	1.75	1.96	2.08
EESAM						
1.5625 µg/mL	28.6	28.2	25.4	0.26	0.2	0.31
3.125 µg/mL	25.1	22.6	20.7	0.38	0.42	0.4
6.25 µg/mL	20.5	19.8	18.1	0.38	0.45	0.58
12.5 µg/mL	15.4	11.3	12.6	0.78	0.69	0.89
25 µg/mL	10.2	9.6	8.9	1.35	2.39	3.06
FDSAM						
1.5625 µg/mL	27.9	24.3	26.8	0.38	0.29	0.31
3.125 µg/mL	24.9	22.8	21.2	0.4	0.42	0.53
6.25 µg/mL	19.6	20.2	16	0.38	0.49	0.48
12.5 µg/mL	9.9	8.9	8.1	0.88	0.79	1.02
25 µg/mL	7.1	4.9	6.9	0.98	1.51	2.35

Source: Authors, 2022. * Not observed during this period.

**Table 3 molecules-28-00231-t003:** Prediction of possible targets for Annonacin.

Order	Code PDB	Target Name	Fit Score
1	5P21	GTPase HRas	5.24
2	1RLB	Transtirretina	4.77
3	1CBS	Cellular retinoic acid binding protein (CRABP2)	4.72
4	1R5L	Alpha-tocopherol transfer protein	4.49
5	1O1V	Ileal lipid binding protein (ILBP)	4.39
6	1SR7	Progesterone receptor	4.37
7	1M7Q	Mitogen-activated protein kinase (MAPK14)	4.37
8	1BOA	Methionine Aminopeptidase 2	4.32
9	1GNI	Seroalbumin	4.30
10	1V4S	Glucokinase	4.30

Source: Authors.

**Table 4 molecules-28-00231-t004:** Annonacin toxicity prediction.

ADMETLab	
hERG	Negative
hepatotoxicity	Negative
skin sensitization	Negative
Reduction of liver damage	Negative
AMES	Negative
PreADME	
Algae	Toxic
*Medaka* sp.	Very toxic
*Daphnia*	Toxic
*Minnow*	Toxic
Ames	Negative
Carcinogenic (rats)	Negative
Carcinogen (mice)	Positive
hERG	Negative
Osiris	
Mutagen	Negative
Tumorigenic	Negative
Irritating	Negative
Effect on reproduction	Negative
ProTox	
Hepatotoxicity	Negative
Carcinogenicity	Negative
Immunogenicity	Positive
Mutagenicity	Negative
Cytotoxicity	Positive

Source: Authors.

## Data Availability

Data are available from the corresponding author upon request.

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
