# Peer review of "Evaluation of Genotoxicity and Toxicity of Annona muricata L. Seeds and In Silico Studies"

_molecules, 2022, doi:10.3390/molecules28010231_

Round 1

Reviewer 1 Report

1. Introduction needs to be improved with detailed phytochemicals and medicinal uses of the plants.

2. Fig.01; TLC profiling chromatogram is very much unclear and no significant conclusion can be made.

Fig 05 & 08; Required clear view.

3. Spectroscopic part is highlighted more than the other experimental validation.

4. In silico work done are very basic in nature.

Author Response

Response to Reviewer 1 Comments

Greetings!

Dear reviewer,

Below are our considerations.

1) Introduction needs to be improved with detailed phytochemicals and medicinal uses of the plants.

Changes were made.

2) Fig.01; TLC profiling chromatogram is very much unclear and no significant conclusion can be made. Fig 05 & 08; Required clear view.

We agree that the TLC profile is not clean, however, the use of Kedde reagent as specific for acetogenins, since it reacts with the y-lactone unit producing a pink color reaction (Grzybowski, 2011; Bruginsk, 2016; Silva, 2017; Lara, 2018) allowed us to infer that it is present in the profile, thus offering significant data that led us to the other tests performed. Fig. 05 & 08 were revised.

3) Spectroscopic part is highlighted more than the other experimental validation.

More information was added to the in silico methods, seeking to demonstrate how this method was constructed and used in this work. The dedication to elucidating the chemical profile of the extract and the fraction is in line with the guidelines of the journal, that it will not accept papers in which the chemical elucidation is faulty or incomplete, therefore, we seek, as much as possible, to carry out tests that contribute to this objective. In addition, chemical elucidation is fundamental in our central hypothesis, in which there is a direct relationship between the toxicity of A. murricata and specific groups of bioactives.

4) In silico work done are very basic in nature.

We understand your position, however, the studies expressed here allowed us to collaborate with the field of studies that have acetogenins as targets for new therapies, mainly antineoplastic. By subjecting Annonacin to different theoretical models to assess its toxicity, we obtained data that support our central hypothesis that acetogenins interfere with neoplastic processes. In addition, it offers data that, in the future, can be used in in vitro and in vivo studies, ensuring that, when chosen, acetogenin will offer efficacy and safety in its use, thus being able to be taken from a hit to lead. Meanwhile, the molecular docking study allowed us not only to collaborate with the hypothesis of the mechanism of action for acetogenins-Complex I (where the formation of hydrophilic and hydrophobic bridges is necessary) but also allowed us to identify which are the pharmacophoric groups that contribute to this activity, contributing to the rational planning of drugs, where we can have antineoplastic drugs that are more active and less toxic.

Hoping you are well, we look forward to hearing from you.

Yours sincerely.

Reviewer 2 Report

The introduction is much short. Please add more reference about genotoxicity and toxicity of natural product or bioactive compounds or in silico researches. 

such as 10.1590/1519-6984.248022 10.1590/1519-6984.244127 10.3390/nu14204426

In the mantext, the 1st, 2nd, 3rd, 7th adn 8th target are expressed in p. 228-241. what about others? Please explain more specificly.

EESAM shows greater deleterious effects on cell division and microcrustaceans. What is your next study about EESAM?

Conclusion should be after Materials and methods

Author Response

Response to Reviewer 2 Comments

Greetings!

Dear reviewer,

Below are our considerations.

1) The introduction is much short. Please add more reference about genotoxicity and toxicity of natural product or bioactive compounds or in silico researches.  Such as 10.1590/1519-6984.248022 10.1590/1519-6984.244127 10.3390/nu14204426

The changes were made, however, the references given as an example were not used. We performed an overview regarding the cytotoxicity of A. muricata and how the studies developed in our work can contribute to offer more robust evidence.

2) In the mantext, the 1st, 2nd, 3rd, 7th adn 8th target are expressed in p. 228-241. what about others? Please explain more specificly.

These targets were chosen because they are related to the carcinogenicity process, whether by proliferation or suppression. In this sense, when the in silico study was carried out, we focused on discussing them, fundamentally, maintaining the central objective of the work, which is to evaluate the relationship between A. muricata and its toxicity. However, we would like to know what you suggest: Remove them? Keep them. We've added an explanation for choice, but we don't know if that satisfies your question about it.

3) EESAM shows greater deleterious effects on cell division and microcrustaceans. What is your next study about EESAM?

We would like to evaluate EESAM in other in vitro studies, such as the micronucleus and comet assays. From the results obtained, perhaps progress to more complex organisms (rats, mice).

4) Conclusion should be after Materials and methods

Answered. Thank you for directing us to the order of topics, we saw several articles, with different formats and we didn't know which one was the cleanest.

Hoping you are well, we look forward to hearing from you.